# GLP-1R Agonists and Their Therapeutic Potential in Inflammatory Bowel Disease and Other Immune-Mediated Inflammatory Diseases, a Systematic Review of the Literature

**DOI:** 10.3390/biomedicines13051128

**Published:** 2025-05-06

**Authors:** Lena Thin, Wei Ling Teh

**Affiliations:** 1Department of Gastroenterology, Fiona Stanley Hospital, Murdoch, WA 6150, Australia; wei.teh@health.wa.gov.au; 2Internal Medicine, Medical School, University of Western Australia, Perth, WA 6009, Australia

**Keywords:** glucan-like peptide-1 receptor agonist, inflammatory bowel disease, Crohn’s disease, ulcerative colitis, obesity

## Abstract

**Background/Objectives:** GLP-1 receptor agonists (GLP-1RAs) have revolutionized weight loss and shown anti-inflammatory actions in several experimental models of colitis. There has been a wealth of recent data suggesting that GLP-1RA treatment may modify disease outcomes in inflammatory bowel disease (IBD). The aim of this systematic review is to determine if GLP-1RAs can act as a sole or adjunctive agent to induce steroid-free clinical remission in IBD patients. **Methods:** The PubMed/Medline, Cochrane Clinical Trial, and EMBASE databases were interrogated with a pre-defined search strategy and eligibility criteria to examine the evidence regarding GLP-1RAs’ use in IBD and non-IBD immune-mediated inflammatory disease (IMID) patients. **Results:** While there is a wealth of pre-clinical animal data suggesting that GLP-1RAs can ameliorate experimental colitis, there is a lack of prospective clinical studies on treating active IBD with GLP-1RAs to specifically induce steroid-free clinical remission. Surrogate data on better IBD composite outcomes have been reported with the use of GLP-1RAs, including a lower risk of surgery, hospitalization, corticosteroid use, and/or the initiation of advance therapies. Data from non-IBD IMID patients are only available for the effect of these agonists on psoriatic plaques, with positive signals. **Conclusions:** The current evidence for the role of GLP-1RAs as a potential anti-inflammatory therapy in IBD is limited, but provides the impetus for much-needed prospective studies and RCTs that include patients with active IBD.

## 1. Introduction

The inflammatory bowel diseases (IBDs), including Crohn’s disease (CD) and ulcerative colitis (UC), are chronic conditions characterized by a disrupted intestinal epithelial barrier and persistent and excessive gastrointestinal inflammation driven by a dysregulated immune response in a genetically susceptible host [1]. Current advanced therapies for moderate to severe IBDs such as biologic agents (anti-TNFα, anti IL12/23, anti-IL-23, and anti-integrins) and small molecules (JAK inhibitors and anti-S1P) have improved the therapeutic armamentarium available to IBD physicians, however, their use is limited by adverse effects, high costs, and a ceiling of effectiveness. As a result, an increasing number of IBD patients experience multiple drug failures, repeated surgeries, disability, and a reduced quality of life. The growing global prevalence of IBD [2] highlights the urgent need to discover new pathways and molecular targets to broaden the therapeutic arsenal available.

Like IBD, the exponential trend of the global obesity epidemic is alarming, with 15–40% of IBD patients being obese [3]. Obesity negatively impacts IBD clinical outcomes, leading to more frequent disease flares, a higher risk of post-operative recurrence, an increased likelihood of developing stricturing or penetrating complications, and an increased risk of losing response to biologics [3,4,5,6,7,8,9,10,11]. There is a lack of high-quality prospective data on whether reversing obesity could improve these outcomes in IBD patients, however, research on psoriasis (PsO) patients suggests that weight loss can effectively reduce psoriasis disease activity [12], along with bariatric surgical interventions [13]. Glucagon-like peptide-1 (GLP-1) is a gut-derived incretin hormone secreted by intestinal L-cells, the alpha cells in the pancreas and the central nervous system [14], in response to nutrient intake. Synthetic GLP1 receptor agonists (GLP-1RAs) have recently revolutionised weight loss in obese patients. Beyond its established roles in enhancing insulin secretion, inhibiting glucagon release, and promoting satiety, GLP-1 exerts pleiotropic immune modulatory and anti-inflammatory effects in multiple systems such as the renal, cardiovascular, and central nervous systems [15]. Additionally, it has anti-inflammatory, cytoprotective, and pro-regenerative effects on the gastrointestinal tract [16]. The selective use of GLP-1 RAs in overweight IBD patients may, therefore, offer a novel therapeutic approach, independent of their role in causing weight loss. This systematic review aimed to evaluate the evidence on GLP-1RAs’ immune-modulating role in IBD and other IMIDs. We primarily examined if GLP-1RAs can achieve corticosteroid-free clinical remission without altering the underlying immunosuppressive therapy in IBD, and secondarily if GLP-1RA therapy can improve IBD outcomes such as reducing corticosteroid use, hospitalizations, surgeries, and advanced therapies. Additionally, we assessed whether GLP-1RAs can improve disease activity in non-IBD IMIDs.

## 2. Materials and Methods

The study was registered on the PROSPERO international prospective register for systematic reviews (registry ID: 1022122). The PRISMA 2020 guidelines [17] were followed for the correct reporting of this systematic review.

### 2.1. Search Strategy

We searched the PubMed/MEDLINE, EMBASE, and the Cochrane Library databases from inception to 31 January 2025. The search terms included the following: (glucagon like peptide 1 receptor agonist) or (glucagon-like peptide 1 receptor agonist) or (glucagon like peptide-1 receptor agonist) or (glucagon-like peptide 1 receptor agonist) or (GLP1 receptor agonist) or (GLP 1 receptor agonist) or (GLP-1 receptor agonist) or (GLP-1 receptor analogue) or (glucagon like peptide 1 analog*) or (glucagon-like peptide 1 analog*) or (glucagon like peptide-1 analog*) or (glucagon-like peptide 1 analog*) or (GLP 1 analog*) or (GLP-1 analog*) or (GLP1Ra) or (GLP 1Ra) or (GLP-1Ra) or (liraglutide) or (exenatide) or (lixisenatide) or (semaglutide) or (albiglutide) or (dulaglutide) or (tirzepatide) AND (“Crohn’s Disease” [Mesh] OR “Ulcerative colitis” [Mesh] OR “Inflammatory Bowel Disease” [Mesh] OR “Psoriasis”[Mesh] OR”, “Psoriatic Arthritis”[Mesh] OR “Arthritis, Rheumatoid”[Mesh] OR “Spondylitis”[Mesh] OR “Spondylitis, Ankylosing”[Mesh] OR “Spondylarthritis”[Mesh] OR “Spondyloarthropathies”[Mesh]). The search was limited to articles published in the English language and original research articles.

### 2.2. Selection Criteria

Review articles were excluded, but their references were examined for eligible articles. Abstracts from conference proceedings and duplicate articles were also excluded, with the paper containing a greater number of patients being included. Case reports were omitted, and only case series with more than two patients were incorporated. For IBD, basic science studies using animal models or in vitro studies in humans evaluating the effects of GLP-1 RAs on the bowel were included, as well as clinical studies assessing GLP-1 RAs in patients with IBD. For studies on IMID patients, only in vitro and clinical studies in human patients were included; animal studies were excluded from this section, as they were beyond the scope of this review. Additionally, studies focusing exclusively on the metabolic or cardiovascular effects of GLP-1 RAs, editorials, opinion pieces, and letters to the editor were excluded, except for checking references for eligibility. Studies that focused exclusively on DPPV-IV inhibitors and on outcomes other than IBD and IMIDs were excluded. The titles and abstracts for IBD studies (author: WT) and IMID studies (author LT) were screened for the pre-defined eligibility criteria, and both authors determined the inclusion of contentious studies by consensus discussion.

### 2.3. Data Extraction and Assessment of Study Quality

For animal and human in vitro studies, author WT extracted the experimental model, study design, and outcome measures for preclinical experimental colitis studies, while author LT did so for non-IBD IMID human in vitro studies. No assessment of study quality was performed for preclinical and in vitro studies. For human clinical studies in IBD (by author WT) and non-IBD IMIDs (by author LT), data on setting, study design, participant information, intervention, comparators, and outcomes were reported using pre-specified structured data collection tables. Both authors graded the quality of the human clinical studies using the Joanna Briggs Institute scoring system [18] for the relevant study type.

## 3. Results

Figure 1 shows that, out of 367 research articles, 42 duplicates were removed. After further exclusions, the following 31 studies were eligible: 20 on IBD, 9 on preclinical animal studies, and 11 on human clinical studies. For non-IBD IMIDs, there were three human in vitro studies and eight human clinical studies.

The results will be presented in the following three sections: Section 1: Evidence from animal studies on the impact of GLP-1 RAs on intestinal inflammation in experimental colitis; Section 2: Evidence from human clinical studies on the effects of GLP-1 RAs in modulating IBD outcomes; Section 3: Evidence from human in vitro and clinical studies on the effect of GLP-1 RAs on disease activity in non-IBD immune-mediated inflammatory diseases (IMIDs).

### 3.1. Evidence from Animal Studies on the Impact of GLP-1 RAs on Intestinal Inflammation in Experimental Colitis

GLP-1RAs, including liraglutide, dulaglutide, exendin-4, and GLP-1 nanomedicine, have demonstrated significant potential in alleviating colonic inflammation across multiple experimental models. Appendix A summarizes the detailed outcomes of the nine experimental studies that were included in this section of the systematic review, which examines the effect of GLP-1RAs on experimental colitis.

#### 3.1.1. Effect on Anti-Inflammatory Cell Signalling and Cytokine Expression

Five studies examined the beneficial effects of GLP-1RAs on macroscopic and microscopic changes in colitis [19,20,21,22,23] and six studies examined their effects on the reduction in proinflammatory cytokines [19,20,21,22,23,24]. These benefits appear to be mediated by increasing the cAMP levels on intraepithelial lymphocytes (IELs) [20] or colon smooth muscle cells (CSMCs) [24] and/or by enhancing the function of Group 3 Innate Lymphoid cells (ILC3) [19]. Several models also show that GLP-1RAs modulate the NFκB- [21,24] and protein kinase A (PKA)-dependent [25] pathways.

Sun et al. investigated the effects of liraglutide compared to a control in a DSS-induced mouse model of colitis [19]. Liraglutide treatment led to slower body weight loss, an extended colon length, and a lower disease activity index (DAI). Additionally, liraglutide-treated mice exhibited an improved histological score and reduced goblet cell loss. However, beneficial effects on colon length were absent in Rorc ^gfp/gfp^ mice, which lack ILC3s, suggesting that liraglutide’s protective effects in colitis are dependent on ILC3s. Flow cytometry analysis after liraglutide treatment revealed a significant reduction in eosinophil count, while neutrophil levels remained unchanged. Moreover, the proportion of IL-22-producing ILC3s was significantly higher in liraglutide-treated mice, whereas GM-CSF and IL-17 expression remained unchanged, serum IL-22 levels (a vital cytokine involved in intestinal epithelial homeostasis, regeneration, and repair) were elevated, and IL-1β expression remained unchanged. In sum, liraglutide was shown to promote IL-22 production by ILC3 without increasing proinflammatory cytokines, leading to improved inflammation in the DSS-induced colitis model.

In another DSS-induced colitis model, Yusta et al. investigated the effects of Exendin-4 (Ex-4) on cytokine expression in IELs [20]. mRNA expression was analysed using real-time quantitative PCR (RT-qPCR), while protein expression was assessed with a Cytometric Bead Array assay kit. The study found that Ex-4 significantly reduced both the mRNA and protein expression of the pro-inflammatory cytokines IL-2, IL-17a, IFNγ, and TNFα in IELs activated by immobilized anti-CD3 and soluble anti-CD28 antibodies. Additionally, Ex-4 markedly upregulated several immunomodulatory genes (Il1b, Il6, Il22, Il12b, TNFα, Ccl2, Cxcl1, and Cxcl2) and antimicrobial genes (RegIIIγ and RegIIIβ). The expression of IL-5 and IL-13, which are critical for pathogen clearance, was also enhanced. These findings suggest that Ex-4 plays a dual role in modulating the intestinal immune response by suppressing pro-inflammatory cytokine production while simultaneously promoting immunomodulatory and antimicrobial defence mechanisms in the intestine.

Mahdy et al. demonstrated the anti-inflammatory properties of dulaglutide using an acetic-acid-induced colitis model in rats [21]. Dulaglutide administration at doses of 50, 100, and 150 μg/kg significantly reduced colon weight, the weight-to-length ratio, both macroscopic and microscopic colonic changes in inflammation, and decreased serum CRP and LDH levels. Immunohistochemistry analysis showed that the addition of dulaglutide reduced colonic NF-κB and IL-6 expression, while spectrophotometric analysis showed a reduction in colonic PI3K, AKT, IFN-γ, and TGF-β1 expression. TGF-β, in turn, is known to rapidly activate PI3K, leading to a significant increase in AKT phosphorylation, while phosphorylated AKT (p-AKT) activates NF-κB, thereby promoting the exponential production of inflammatory cytokines. These findings suggest that dulaglutide modulates inflammatory signalling pathways via its effects on TGF-β1, PI3K/AKT, NF-κB, IL-6, and IFN-γ production.

Al-Dwairi et al. showed that in mouse colon smooth muscle cells stimulated with lipopolysaccharide (LPS), exendin-4 significantly reduced the expression of inflammatory cytokines and chemokines, including TNF-α, IL-1α (a central mediator of innate immunity and inflammation), TCA-3 (chemotactic factor for neutrophils and macrophages), SDF-1 (chemotactic factor for lymphocytes and monocytes), and M-CSF (controls the survival, proliferation, and differentiation of macrophages), as analysed via antibody array membrane, ELISA, and real-time PCR [24]. LPS increased NF-κB phosphorylation, which was abrogated by the addition of exendin-4. Additionally, Exendin-4 was shown to increase cAMP levels, which is a negative regulator of the CREB/PKA-dependent T cell signalling pathway [24].

Exendin-4 has been shown to induce CREB (cAMP response element-binding protein) phosphorylation, a surrogate marker of Protein Kinase A (PKA) activity in T cells. Experiments by Wong and colleagues [25] showed that GLP1R agonism on IELs inhibited proximal TCR signalling in a PKA-dependent manner [25] and decreased the IEL production of IFNγ and TNFα, thereby abrogating T-cell mediated gut epithelial inflammation and destruction. In a T-cell-driven adoptive transfer (AdTr) colitis mouse model, liraglutide significantly improved disease activity markers, including histological activity in colonic tissue and the colon weight-to-length ratio, likely due to its ability to reduce CCL20, IL-33, and IL-22 [23].

#### 3.1.2. Effect on Gut Microbial Homeostasis

In a DSS colitis mice model, Sun et al. demonstrated liraglutide’s effect on modulating the gut microbiota by performing 16S rRNA sequencing on faecal samples [19]. At the phylum level, *Firmicutes* and *Proteobacteria* were increased, with *Firmicutes* being negatively correlated with gastrointestinal inflammation. At the genus level, there were increased abundances of *Lactobacillus*, *Helicobacter*, *Turicibacter*, and *Alistipes*, while *Staphylococcus* and *Faecalibaculum* were reduced. Species-level analysis further identified an elevated presence of *Lactobacillus Reuteri*, *Lactobacillus Johnsonii*, and *Helicobacter typhi*, microbial species known to enhance IL-22 expression in ILC3s.

Metabolites within the gut play a crucial role in mediating interactions between microbiota and the host, contributing to intestinal homeostasis. Comparative metabolomics analysis revealed that GLP-1RAs significantly altered both faecal and colonic metabolic profiles. Pathway enrichment analysis identified sphingolipid signalling and sphingolipid metabolism as key pathways influenced by GLP-1RAs. Notably, GLP-1RA treatment increased the relative abundance of N, N-dimethyl sphingosine (DMS), an endogenous metabolite derived from sphingosine. DMS functions as a sphingosine kinase inhibitor, preventing the conversion of sphingosine into sphingosine-1-phosphate (S1P). S1P is primarily associated with pro-inflammatory effects, as it binds to S1PR1 receptors on lymphocytes, facilitating their migration to inflamed tissues. DMS administration significantly alleviated colitis, as indicated by an increased colon length, reduced serum IL-6 levels, and improved histological scores. Additionally, DMS treatment led to a reduction in monocyte and macrophage populations while increasing the absolute number of ILC3s, thereby enhancing IL-22 production. These findings suggest that DMS may act as a key metabolite generated by a GLP-1RA-modulated microbiome, contributing to increased IL-22-producing ILC3s and the subsequent amelioration of intestinal inflammation [19].

#### 3.1.3. Effect on Gut Barrier Function

Lymphotoxin beta receptor (LTβR) signalling in intestinal epithelial cells (IELs) is critical for epithelial IL-23 production, while IL-18 serves as a key IL-22-regulated gatekeeper of the intestinal epithelial barrier. Sun et al., with RT-PCR studies on Rag1^−/−^ DSS colitis mice (which lack T and B cells) treated with liraglutide, demonstrated the significant upregulation of the following IL-22-dependent genes: Ltbr, Il18, and RegIIIβ, while the expression of Glp1r, Tjp1 (encoding the tight junction protein ZO-1), and Il23 remained unchanged. Immunohistochemistry further confirmed the increased expression of claudin-1 and ZO-1, indicating enhanced tight junction integrity in the intestinal epithelial layer [19].

In the acetic-acid-induced colitis model, dulaglutide enhanced intestinal barrier function, upregulating colonic Trefoil Factor (TFF)-3 expression, as measured by enzyme-linked immunosorbent assay (ELISA) [21]. TFF-3 strengthened the intestinal barrier by regulating tight junctions, thereby reducing paracellular permeability. Primarily co-expressed with MUC2 in intestinal goblet cells, TFF-3 plays a crucial role in mucosal regeneration and repair. In Brunner’s glands, GLP-1 has been shown to induce the production of IL-33 (associated with both proinflammatory and anti-inflammatory functions), CCL20 (related to intestinal inflammation and mucosal healing), and MUC5B, which are essential for epithelial defence mechanisms [23].

In contrast, Kato et al. proposed that liraglutide increased *E. coli* levels in mice via sympathetic nervous system activation, as well as attenuating tight junction gene expression in the caecum of DSS colitis mice and, hence, promoting bacterial translocation [26]. Caseinolytic protease B (ClpB), a protein component of *E. coli*, is significantly correlated with *E. coli* 16S rRNA expression. Liraglutide administration led to a striking 400-fold increase in ClpB expression compared to the control. Additionally, liraglutide significantly increased norepinephrine (NE) levels in both plasma and caecal contents. However, when liraglutide was co-administered with medetomidine, an α2 receptor agonist that suppresses sympathetic nervous system (SNS) activity, neither caecal ClpB expression nor plasma and caecal NE levels increased. This suggests that liraglutide induced a surge in *E. coli* by stimulating SNS activity and enhancing NE release into the intestinal lumen. In the DSS colitis mouse model, liraglutide-treated mice exhibited significantly lower caecal occludin mRNA levels and higher TNF-α mRNA levels compared to controls, while RegIIIβ and IL-33 mRNA levels showed an increased trend. The study also investigated bacterial translocation (BT) by detecting *E. coli* translocation via PCR and showed that liraglutide-treated mice had significantly higher BT rates compared to controls. Furthermore, caecal occludin mRNA levels were notably lower in BT-positive mice compared to BT-negative ones. Collectively, these findings suggest that liraglutide promotes bacterial translocation by impairing intestinal barrier integrity through the downregulation of occludin, a key tight junction gene. This result, contrasting prior findings of increased barrier integrity, requires further exploration.

#### 3.1.4. Effect on Oxidative Stress

Mahdy et al. also demonstrated that dulaglutide pretreatment can effectively mitigate oxidative stress, as indicated by increased levels of antioxidant markers such as serum TAC (total antioxidant capacity), colonic SOD (superoxide dismutase), and GSH (glutathione), alongside reduced MDA (malondialdehyde) concentrations [21]. MDA serves as a key indicator of oxidative stress due to its role in lipid peroxidation caused by excessive free radical production. The body’s natural defence system against oxidative stress relies on antioxidant enzymes like SOD and GSH, with TAC serving as a broader measure of antioxidant capacity.

#### 3.1.5. Other Effects

Biagioli et al. proposed that targeting the GPBAR1/GLP-1/ACE2 axis could offer a promising therapeutic strategy for IBD [27]. GPBAR1, also known as TGR5, is a bile acid receptor highly expressed in the intestinal epithelium of both the ileum and colon, as well as in L cells, endothelial cells, neurons, and myeloid-derived immune cells. In the small intestine, GPBAR1 activation stimulates the secretion of GLP-1. Meanwhile, ACE2—a carboxypeptidase that converts Ang II to Ang-(1-7)—plays a role in regulating dietary amino acid homeostasis, innate immunity, and gut microbial balance. The authors showed that colonic ACE2 expression increased during inflammation, as shown by elevated ACE2 levels in colonic tissue samples from Crohn’s disease patients. The GPBAR1 agonist BAR501 was shown to protect against oxazolone-induced colitis. This protection was evident from improvements in the Crohn’s Disease Activity Index (CDAI), colonic macroscopic features, endoscopic findings, and a reduced expression of pro-inflammatory cytokines in the colon. Additionally, BAR501 upregulated ACE2 expression by about 30% compared to oxazolone-treated controls and significantly increased the expression of Gcg, the gene encoding GLP-1. Liraglutide did not interfere with the anti-inflammatory effects of BAR501, as indicated by stable IL-8, IL-1β, and CCL2 mRNA levels in HT29 cells (human colorectal adenocarcinoma cells), however, liraglutide did enhance ACE2 mRNA expression in co-cultures of HT29 and U937 cells (human monocyte cells) stimulated with LPS, thereby demonstrating its anti-inflammatory action.

In summary, GLP-1RAs exhibit a broad range of anti-inflammatory, gut-barrier-protective, and microbiome-modulating effects in experimental colitis. Their multifaceted role in gut homeostasis underscores their potential as a therapeutic option for IBD.

### 3.2. Evidence from Human Clinical Studies on the Effects of GLP-1 RAs in Modulating IBD Outcomes

A total of ten studies met the inclusion criteria, comprising six cohort studies [28,29,30,31,32,33], two case series [34,35], and two case–control studies [36,37]. These studies were retrospective in design, covering both diabetic and non-diabetic (overweight/obese) populations with CD and UC. There were no prospective randomized controlled trials. The sample sizes ranged from 16 to 68,443 patients, with study durations varying between 6 months and 12 years. No studies specifically reported on the proportion of patients that achieved steroid-free clinical remission at any time point as an outcome, as no studies directly assessed the effect of treating IBD patients with active inflammation with GLP-1RAs either as a sole agent or as an adjunct to existing IBD therapies. Thus, most studies either reported improvements in composite outcomes (or a lack of benefits) or any flares of the underlying IBD activity. No studies reported the harmful effects of GLP-1RAs on IBD course. Table 1 summarizes the key findings from these ten studies.

In accordance with the results of preclinical studies, several clinical studies have suggested positive effects of GLP-1RAs on IBD course. Villumsen et al. conducted a nationwide Danish registry study involving 3751 patients with IBD and type 2 diabetes mellitus (T2DM), comparing those receiving GLP-1-based therapies (GLP-1 receptor agonists and DPP-4 inhibitors) to those on other antidiabetic medications (*n* = 2769) [28]. There was a 48% lower incidence of composite IBD-related outcomes (defined as the need for oral corticosteroids, TNF-α inhibitors, IBD-related hospitalization, or surgery) among patients on GLP-1-based therapies (adjusted incidence rate ratio [IRR]: 0.52, 95% CI: 0.42–0.65). When evaluating individual components of the composite outcome, both oral corticosteroid use (adjusted IRR: 0.54, 95% CI: 0.41–0.70) and IBD-related hospitalization (adjusted IRR: 0.73, 95% CI: 0.58–0.91) were significantly lower amongst GLP-1 RA/DPP-4 inhibitor users. While the risk of initiating TNF-α inhibitors (adjusted IRR: 0.56, 95% CI: 0.32–1.00) and IBD-related surgery (adjusted IRR: 0.79, 95% CI: 0.57–1.09) was also reduced, these associations were not statistically significant. Sensitivity analyses focusing exclusively on GLP-1 RAs showed consistent results, with a 44% reduction in the risk of reaching the composite IBD outcome (adjusted IRR: 0.56, 95% CI: 0.39–0.83). Similarly, Desai et al. conducted a large cohort study involving 2270 patients with IBD and T2DM, demonstrating that GLP-1RA use was associated with a 63% reduction in the risk of colectomy among UC patients (aHR: 0.37, 95% CI: 0.14–0.97) and a 45% lower risk of IBD-related surgery in CD patients (aHR: 0.55, 95% CI: 0.36–0.84) [30]. There were no significant differences observed for the risk of intravenous steroid use (UC (aHR: 1.21, 95% CI: 0.92–1.59); CD (aHR: 1.04, 95% CI: 0.80–1.34), the likelihood of needing oral corticosteroids (aHR: 1.12, 95% CI: 0.96–1.31), or the need to initiate any advanced therapies (aHR: 0.82, 95% CI: 0.51–1.33). Gorelik et al. recently analysed a cohort of 3737 patients with IBD and T2DM, demonstrating that the use of GLP-1RAs was associated with a 26% reduction in the risk of composite IBD outcomes, defined as steroid dependence, the initiation of advanced IBD therapy, hospitalization, surgery, or death (aHR: 0.74, 95% CI: 0.62–0.89) [32]. While similar trends were observed in the multivariate analyses of individual components, only hospitalization as an outcome was associated with a statistically significant reduction (aHR: 0.74, 95% CI: 0.61–0.91). The protective effect of GLP-1 RAs was particularly evident in patients with obesity (aHR: 0.61, 95% CI: 0.50–0.77), whereas no significant benefit was noted in non-obese individuals (aHR: 0.94, 95% CI: 0.67–1.31).

Other studies have reported neutral associations between GLP-1RA use and IBD outcomes, with no disease worsening, but also no significant benefit to the course of IBD. Ramos Belinchón et al. evaluated 16 obese patients with IBD treated with semaglutide or liraglutide and observed no significant deterioration in IBD activity [34]. While two patients experienced clinical worsening, only one required oral budesonide. No patients required hospitalization, surgery, the escalation of biologic therapy, or systemic corticosteroids. Desai et al. conducted a population cohort study involving 47,424 obese IBD patients and compared 150 IBD patients treated with semaglutide to an IBD cohort that did not receive semaglutide or other anti-obesity medications (AOMs) [29]. After propensity score matching, there was no significant difference found for the risk of oral steroid use (aOR: 0.81, 95% CI: 0.48–1.36), intravenous steroid use (aOR: 0.69, 95% CI: 0.29–1.61), the need to initiate advanced therapy (aOR: 1.03, 95% CI: 0.41–2.60), or the risk of any-cause emergency department visits (aOR: 0.92 (0.52–1.61) at one year between the semaglutide-treated and non-semaglutide-treated IBD patients. The semaglutide-treated IBD patients however, had a lower risk of any-cause hospitalization (aOR: 0.35, 95% CI: 0.19–0.67). Notably, no patients in the IBD semaglutide group required surgery compared to ten that did have surgery in the non-semaglutide-treated IBD group. Anderson et al. conducted a retrospective analysis of 120 IBD patients treated with GLP-1 receptor agonists and found a significant reduction in CRP levels over one year (12.92 vs. 6.38 mg/dL, *p* = 0.005) [35], however, there were no observed differences in IBD-related hospitalizations, clinical disease activity (measured by the Harvey–Bradshaw Index or Modified Mayo Score), or endoscopic scores between the year before and after treatment with GLP-1RAs. Conversely, a cohort study by St-Pierre et al. involving 36 IBD patients who initiated semaglutide or tirzepatide for weight loss found no significant changes in CRP levels in the 25 patients that had repeated CRP measurements (median baseline CRP was 3 mg/L (IQR: 3–6.5 mg/L), which remained unchanged at the end of follow-up [31]. Levine et al. examined 224 IBD patients on GLP-1 receptor agonists and found no significant change (*p* = 0.36) in the rates of IBD exacerbation (measured as a composite of IBD-related hospitalizations, steroid prescriptions, medication escalation or adjustments, or IBD-related surgeries) after 12 months of GLP-1 RA therapy, compared to the 12 months prior to treatment [36]. Pham et al. compared the use of anti-obesity medications (AOMs), a subset of which were GLP-1 receptor agonists, in 36 IBD patients compared to 36 non-IBD controls and found no significant impact on the frequency of IBD exacerbations (which occurred in 7 patients, 19.4%), defined as the need for steroid use, a change in IBD therapy, hospitalization, and surgery compared to pre-treatment status [37].

Weight loss was a consistent outcome across these studies [31,34,35,36,37]. Desai and Levine reported that weight reductions in GLP-1 RA users were comparable to those observed in non-IBD matched controls [29,36], with greater reductions observed with Tirzepatide [29]. GLP-1 RAs were well tolerated, with most adverse effects limited to mild gastrointestinal symptoms. Additionally, Nielsen et al. found no increased risk of ileus or intestinal obstruction among Danish IBD patients receiving GLP-1 RAs [33].

### 3.3. Evidence from Human In Vitro and Clinical Studies on the Effect of GLP-1 RAs on Disease Activity in Non-IBD Immune-Mediated Inflammatory Diseases (IMIDs)

Only three human (one in PsO and two in rheumatoid arthritis (RA)) in vitro studies met the eligibility criteria for inclusion as basic science evidence that GLP-1RAs have therapeutic anti-inflammatory potential for treating non-IBD IMIDs (Table 2). A Danish study observed that GLP-1R gene expression was detected in five out of six PsO patients in areas with active psoriatic plaques compared to five out of six PsO patients that had no GLP1R expression in inactive areas and five out of six healthy controls (without psoriasis) that also had no expression of GLP1R, suggesting that GLP1R expression is upregulated in active psoriatic lesions [38]. Moreover, upon the stimulation of keratinocyte cell suspensions from these patients with TNFα and γIFN, there was no GLP-1R expression detected, indicating that the expression of GLP1R in psoriatic plaques was from immune cells expressing these receptors. IL-17, a key cytokine in psoriasis pathogenesis, was also found to be highly expressed in psoriatic plaques compared to the skin for healthy controls [38]. Two Chinese studies were conducted in RA patients [39,40]. Both studies examined the effect of GLP-1RA on fibroblast-like synoviocytes (FLSs), cells in the synovial fluid of joints that normally play a protective and homeostatic role in joint function but can transform into a pathological phenotype that is apoptosis-resistant and proinflammatory in function in the presence of inflammatory stimuli. The study by Tao et al. [39] examined the result of stimulating FLS with 10 ng/mL of TNF-α and then the effect of adding increasing doses (10 and 20 nM) of exenatide on FLS function, whereas Du et al. [40] examined the effect of stimulating FLS with 10 ng/mL of IL-1β and the protective sequalae of adding the GLP1RA lixisenatide (also in increasing doses of 10 and 20 nM) on FLS function (Table 2). In both studies, stimuli with TNFα and IL-1β, respectively, led to decreased mitochondrial function, increased oxidative stress, increased levels of reactive oxygen species (ROS), increased expressions of proinflammatory cytokines (TNFα, IL-1β, IL-6, and IL-8) and expressions of matrix-metalloproteinase 1, 3, and 13, and upregulation of the p38/MAPK and p-65/NFκB transcriptional complex expressions of the proinflammatory signalling pathways (Table 2) [39,40]. The addition of GLP-1RAs (exenatide and lixisenatide, respectively) led to the abrogation of these proinflammatory signalling proteins and cytokine expression (Table 2).

Eight clinical studies [41,42,43,44,45,46,47,48] met the eligibility criteria for review to examine the effectiveness of GLP-1RAs in modulating inflammation in the non-IBD IMID population (Table 3). All eight studies were conducted in patients with plaque psoriasis (PsO). In five of these studies [42,43,44,45,46], GLP1RAs were studied in type 2 diabetes mellitus (T2DM) patients. In five studies [41,43,46,47,48], patients were specifically mentioned to be concurrently overweight or obese, defined as a body mass index (BMI) of ≥25 kg/m^2^. Only one study specifically mentioned that patients with psoriatic arthritis (PsA) were included, which comprised 30% of their cohort [41], whereas five studies specifically mentioned the exclusion of patients with PsA [43,44,45,47,48], and the remaining two did not mention the inclusion or exclusion of PsA patients. Four studies were prospective open-label cohort studies [41,43,44,45], with from seven to twenty PsO patients enrolled without a control arm, and one was a prospectively studied case series of three PsO patients [46]. Three studies were randomised controlled studies (RCTs) [42,47,48], among which only one was performed in a placebo-controlled, double-blinded fashion [47]. Five of the eight studies utilised liraglutide exclusively [41,42,43,44,47], two used liraglutide and exenatide (in separate patients) [45,46], and one used semaglutide [48]. Most studies had a study duration of 12 weeks, with a range from 6 weeks to 9 months (Table 3). Single-arm, prospective, controlled studies compared dermatological outcomes after a specified duration of therapy with the baseline status. The three small RCTs evaluated differences in the treatment arm compared with either placebo (while continuing a stable background therapy for PsO) [47] or standard of care treatment for PsO patients (topical keratolytic treatments or oral acitretin capsules) [42,48]. All five non- RCT studies found a significant improvement in mean Psoriasis Area Severity Index (PASI) scores at the end of the treatment period compared to before the treatment started (*p* < 0.05). Three of these five studies [41,43,44] also reported a significant improvement in the Dermatology Life Quality Index (DLQI) score at the end of treatment compared to baseline (*p* < 0.05). Additional in vitro work by some of these groups might explain how GLP-1RAs modulate skin inflammation in PsO patients. Hogan et al. [46] found that T (iNKT) cells, a rare subset of innate T cells that have immune regulatory properties, expressed GLPRs, and that there was a dose-dependent inhibition of GLP-1 on the iNKT cell cytokine secretion of γIFN and IL-2. The same group also showed that giving GLP1RAs increased cAMP levels, which, in turn, activated the CREB transcription factor for increased IL-10 (an anti-inflammatory cytokine) production. Ahern et al. [43] subsequently found that the % of circulating iNKT cells increased by 37.9% (IQR: 18.5–234.6, *p* = 0.03) upon administration of the GLP1RA. The group also found that the % of TNFα-producing monocytes numerically decreased by 53% ((IQR: 51.4–55.0), *p* = 0.07). Buysschaert et al. [45] found that giving GLP1RAs decreased the infiltration of dermal γδT cells into the skin of PsO patients and that this decrease strongly correlated with the decrease in PASI scores (r = 0.894, *p* = 0.007), whilst the decrease in IL-17 expression did not reach statistical significance.

The two open-label RCTs both showed that the intervention arm (GLP-1RA) resulted in a lower PASI score compared to the control arm [42,48] at the end of the study. Notably, the study by Petkovic-Dabic et al. [48] did show that both arms experienced a statistical improvement in median PASI scores, but the change from baseline was significantly greater in the intervention arm compared to the control arm, and this was also true of the DLQI score (Table 3). Interestingly the intervention arm also saw a significant decrease in CRP (*p* = 0.01) and IL-6 levels (*p* = 0.05) compared to the control arm (*p* = 0.5 and *p* = 0.1 respectively), however, IL-17 was not detected at all in either group, and no other changes in cytokine expression were observed [48]. On the other hand, Lin et al. [42] found that IL-23/IL-17/TNFα expressions in the skin of the intervention arm were all significantly lower than in the control arm (*p* < 0.05). The only double-blinded RCT by Faurschou et al. [47] was reported as a negative study, however, the analysis was a comparison of the *difference* in the change in the PASI score between the intervention (−2.6 ± 2.1) and control arm (−1.3 ± 2.4), which was found not to significantly differ (*p* = 0.23). However, when compared to baseline values, the change in the PASI value was statistically lower in the intervention (*p* = 0.0026) compared to the control group (*p* = 0.14). All studies showed a significantly decreased weight and/or body mass index and/or waist circumference after treatment with GLP1RAs compared to the baseline or control groups for the cohort and RCTs, respectively. No serious adverse events were reported, and most mild side effects were gastrointestinal in nature, including nausea, bloating, constipation, and diarrhoea.

## 4. Discussion

GLP-1 is an incretin hormone that is released by L-cells of the small intestines in response to nutrient intake and promotes insulin release to counteract the hyperglycaemia triggered by the meal. GLP-1 receptor activation also inhibits glucagon release, acts centrally in the brain to suppress appetite (including hedonistic appetite), and acts peripherally to delay gastric emptying, resulting in the successful suppression of calorie consumption [49]. The earliest form of a GLP-1RA was exenatide, which was a synthetic derivation of the peptide exendin-4, which was found in the saliva of the venomous lizard, the Gila monster [50]. Since their earliest form, GLP-1RAs have evolved to become more effective and longer lasting in duration, with the latest versions being liraglutide (once-daily injection), semaglutide (once-weekly injection) [49], and tirzepatide (once-weekly injection), the latter which is an agonist of the glucose-dependant insulinotropic polypeptide (GIP) receptor [51].

Evidence from the rheumatology and dermatology literature suggests that weight loss via behavioural and/or surgical interventions can improve disease activity in those affected by obesity and PsA, RA, or PsO [13,52,53,54,55,56,57]. No comparable prospective IBD studies on the impact of weight loss on disease activity have been conducted. As obesity is an inflammatory state, it is conceivable that weight loss would have an immune-modulatory effect on these IMIDs, however, there is also evidence to suggest that immune modulation occurs even before weight loss transpires in those treated with GLP-1RAs [46,58,59] and bariatric surgery [60], where there is a surge in native GLP-1 levels. These observations indicate that GLP-1RAs may have inherent anti-inflammatory, immune-modifying functions that could be harnessed therapeutically for indications other than weight loss. To the best our knowledge, this systematic review is the first to examine the available published scientific evidence on the therapeutic potential of GLP-1RAs in treating IBD. Additionally, we reviewed the evidence regarding the utilization of GLP-1 receptor agonists (GLP-1RAs) in treating other immune-mediated inflammatory diseases (IMIDs). This was conducted to evaluate their therapeutic potential in the management of inflammatory conditions and to contextualize their application. We presented substantial evidence demonstrating the capacity of GLP-1RAs to alleviate experimental colitis by inhibiting pro-inflammatory signalling pathways through the downregulation of the PI3K/AKT, NFκB, and CREB/PKA dependent signalling pathways. Additionally, GLP-1RAs reduce the expression of several pro-inflammatory cytokines, including IL-2, IL-17a, IL-6, IFNγ, and TNFα [20,21,24,25]. Meanwhile anti-inflammatory cytokine production from ILC-3 cells such as IL-22 is upregulated [19]. Furthermore, enhanced gut microbial homeostasis, barrier function, and reduced oxidative stress have been demonstrated (Appendix A).

There is a translational gap, however, between animal and human studies on GLP-1RA therapy in modulating IBD activity. Only data from population cohort, retrospective cohort, and case–control studies exist, demonstrating the effectiveness of GLP-1RAs in modulating composite IBD outcomes regarding surgery, hospitalization, and the escalation of therapy rather than specifically steroid-free clinical remission, which was our a-priori research question. Case–control studies also focused on the effectiveness of GLP-1RAs in promoting weight loss in IBD patients compared to non-IBD controls, rather than evaluating whether the GLP-1RAs effectively controlled inflammatory disease activity. There are no dedicated prospective studies examining the effect of treating active IBD with GLP-1RAs either as a sole or adjunctive agent, and there is a lack of randomized controlled data. The published evidence suggests that treating IBD patients with GLP-1RAs results in equivalent weight loss compared to non-IBD patients [29] and has a positive effect on IBD composite outcomes, such as the risk of surgery and/or hospitalization and/or the need for corticosteroids and/or the initiation of biologic/advance therapies [28,29,30,32]. The remaining studies we identified (Table 1) either showed that GLP-1RAs did not influence disease activity or did not lead to significant complications such as ileus or pseudo-obstruction [33]. We found the clinical studies on GLP-1RAs to be at a low risk of bias given high JBI scores, however, they lacked the characteristics to answer our primary research question. To further understand the role of GLP-1RAs in treating IBD, IBD patients with active gut inflammation, who also have indications to commence GLP-1RAs such as obesity or type 2 diabetes, need to be studied prospectively and in randomised placebo-controlled trials.

In non-IBD IMIDs, the evidence from small prospective, cohort, and case series, all in patients with plaque psoriasis (PsO), showed significant disease improvement, as assessed by PASI and DLQI scores [41,42,43,45,46], upon the administration of GLP-1RAs (Table 3). Although the quality of the evidence was low, limited by small numbers and a short duration of therapy (mostly less than 12 weeks), the improvement in disease activity and quality of life scores is compelling. Additionally, three small RCTs [42,47,48], also in PsO patients, of which only one was deemed sufficient in quality [47], demonstrated a positive effect of GLP-1RA in ameliorating PsO disease activity. We found no published clinical evidence of GLP-1RA use in other IMID conditions such as rheumatoid arthritis, psoriatic arthritis, or spondyloarthropathies. We also presented human in vitro evidence (Table 2) showing the upregulation of GLP-1 receptor expression in inflamed plaques of PsO patients compared to uninflamed areas and the skin of healthy controls [38] and the downregulation of inflammatory pathways upon the addition of GLP-1RAs in two in vitro studies utilizing FLS cultured from rheumatoid arthritis patients [39,40]. Despite their lower quality and higher risk of bias in the clinical studies conducted on non-IBD IMIDs compared to those on IBD, their findings offer promise and a scientific impetus for initiating similar pilot prospective and randomized controlled studies in patients with IBD.

Our systematic review had a few limitations. Firstly, we excluded case reports, many of which appeared in the non-IBD IMID literature, particularly in psoriasis (PsO) patients. However, we decided that the data from individual case reports were insufficient in quality for inclusion in this review. Another limitation was incomplete or missing data, such as the maximum dose of administered GLP-1RAs. A significant challenge in interpreting the data arose from the heterogeneity in the GLP-1RA doses administered and the variable reporting of these doses. For instance, most non-IBD IMID studies did not reach the maximum prescribed dose of 3 mg for Liraglutide, which may have affected the observed outcomes. Furthermore, the lack of reporting on patient-years of follow-up in population-based studies restricted the interpretation of the long-term efficacy of GLP-1RAs in treating IBD, especially when considering the negative results reported by some studies.

## 5. Conclusions

GLP-1RAs may have therapeutic potential for treating IBD patients, but more research is required. Prospective studies and RCTs should be conducted in IBD patients with active inflammation to verify their immunomodulatory effects and assess whether they can be used therapeutically, either alone or as an adjunctive therapy.

## Figures and Tables

**Figure 1 biomedicines-13-01128-f001:**
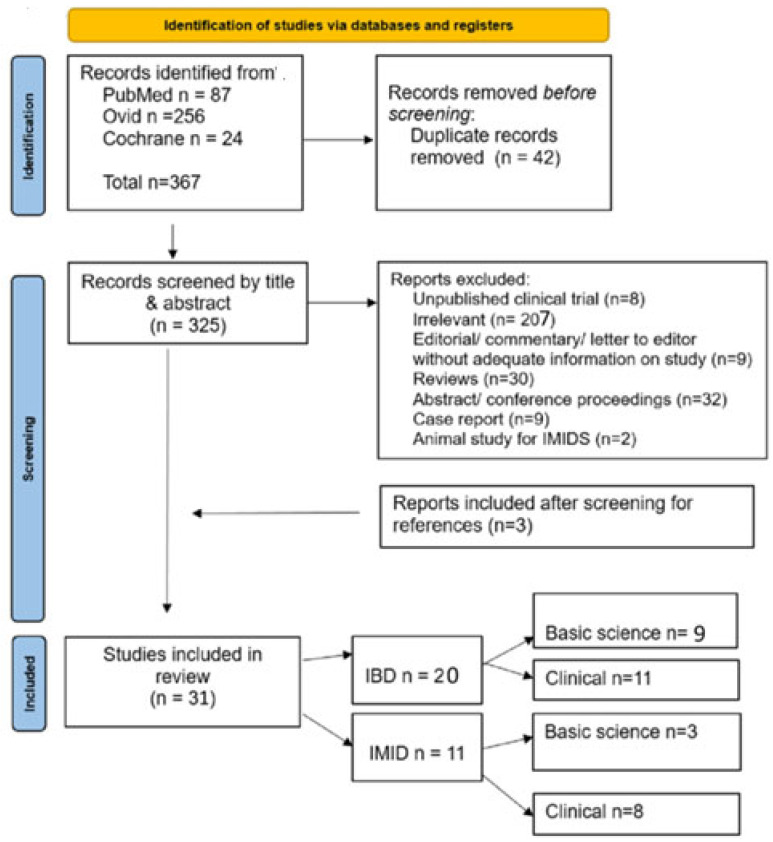
PRISMA flow chart of included and excluded studies.

**Table 1 biomedicines-13-01128-t001:** Human clinical studies of GLP-1RAs’ effect on modulating IBD outcomes.

Author (Year)	Study Origin and Years of Study	Study Type	Patient Cohort	Intervention	Comparator	Major Outcomes Reported	Results		JBI Score Author 1	JBI Score Author 2
Villumsen et al. (2021) [28]	Denmark, 2007–2019	Population cohort	IBD with T2DM n = 3751 CD n = 960, UC n = 2791	GLP-1 based therapies n = 982 (GLP-1RA +/− DPPIV inhibitor)	Conventional anti-diabetic therapy (n = 2769) ^1^	**IBD outcomes:** Composite need for oral corticosteroid treatment, need for anti-TNFα treatment, IBD-related hospitalisation, or IBD-related major surgery **Metabolic outcomes:** Not reported	**IBD outcomes:** Adjusted incidence rate ratio (IRR) for composite outcome: 0.52 (95% CI 0.42–0.65) overall UC: 0.50 (95% CI 0.39–0.65), CD: 0.62 (95% CI 0.41–0.92). IRR for GLP-1 receptor agonists only 0.56 (95% CI 0.39–0.83).		8/11	8/11
Belinchon et al. (2024) [34]	Spain, 2019–2021	Case series	IBD with obesity n = 16 CD n = 9 UC n = 7	Semaglutide (n = 11) or liraglutide (n = 5)	Baseline	**IBD outcomes:** Faecal calprotectin, CRP, disease activity index (CD: decrease in Harvey–Bradshaw index (HBI) of 3 or more points from baseline; UC decrease in partial Mayo score of 2 or more points from baseline), need for IBD therapy escalation **Metabolic outcomes:** % д in body weight and a weight reduction of 5% or more at 6 months	**IBD outcomes:** No significant changes in CRP, faecal calprotectin and disease activity index. One patient required isolated course of oral budesonide **Metabolic outcomes:** д% body weight = −6.2% (−3.4 − [−8.5]). 58.3% (7/12) achieved a 5% or more weight reduction at six months		9/10	9/10
Desai et al. (2024) [29]	USA, 2021–2023	Retrospective cohort study	(i) IBD with obesity, n = 47,424 (ii) non-IBD obese patients on semaglutide, n = 21,019	IBD patients on Semaglutide (n = 150), liraglutide (n = 75), tirzepatide (n = 95)	IBD patients on other anti-obesity medications (AOM) ^2^ (n = 197)	**IBD outcomes:** Risk of oral steroid use, hospitalization requiring IV corticosteroids, initiation of advanced therapies in bio-naïve patients, IBD-related surgery, any-cause hospitalization, and any-cause emergency department (ED) visit **Metabolic outcomes:** Total body weight (TBW) change in pounds from baseline to between 6 and 15 months	**IBD outcomes:** No difference between semaglutide-treated versus non semaglutide/other AOM-treated obese IBD patients for all outcomes apart for lower adjusted OR for any-cause hospitalization (aOR, 0.35; 95% CI, 0.19–0.67) **Metabolic outcomes:** Similar TBW change (*p* = 0.24): −16 ± 13.4 pounds in IBD cohort on semaglutide, −18 ± 12.7 pounds in non-IBD cohort on semaglutide. Semaglutide superior to other AOMs (*p* < 0.01 for all), superior to liraglutide (*p* = 0.04), inferior to tirzepatide (*p* = 0.01) for weight loss		8/11	9/11
Desai et al. (2024) [30]	USA, 2010–2022	Retrospective cohort study	IBD with T2DM UC n = 1130 CD n = 1140	Liraglutide(n = 212), semaglutide (n = 555) or dulaglutide (n = 546)	Conventional oral hypoglycaemics agents ^3^ (UC n = 4615, CD n = 4744)	**IBD outcomes:** Hospitalisation requiring IV methylprednisolone and IBD-related surgery, oral corticosteroid use, first-time advanced therapy initiation within 3 years **Metabolic outcomes:** Not reported	**IBD outcomes:** Lower risk of surgery in GLP-1RA cohort UC aHR: 0.37, 95% CI: 0.14–0.97. CD aHR: 0.55, 95% CI: 0.36–0.84) No significant differences in other outcomes		9/10	8/10
Anderson et al. (2024) [35]	USA, 2014–2024	Retrospective case series	IBD patients on GLP-1RA (n = 120) CD, n = 61 UC n = 59 On GLP-1RA for: Diabetes n = 72, Weight loss n = 43, MASH n = 5	GLP-1RAs Dulaglutide 26 (21.7) exenatide, 8 (6.7) liraglutide 12 (10) semaglutide 74 (61.7) Tirzepatide 0 (0)	Status 1 year prior compared to 1 year after GLP-1RA initiation for IBD outcomes	**IBD outcomes:** Clinical severity scores (HBI for CD, modified Mayo score for UC), Endoscopic scores (Simple Endoscopic Score for CD and the Mayo Endoscopic score for UC), number of IBD-related hospitalizations, and changes in CRP levels **Metabolic outcomes:** % д weight 1 year after GLP-1RA initiation	**IBD outcomes:** Decrease in CRP (12.92 vs. 6.38 mg/dL, *p* = 0.005) No significant differences for other outcomes **Metabolic outcomes**: % д weight = −3.9% (±7.7%)		10/10	9/10
Levine et al. (2024) [36]	USA, 2009–2023	Case–control	IBD patients n = 224 UC n = 97, CD n = 100, unclassified n = 27	GLP-1RAs (n = 224) Semaglutide 148 (66.1) Liraglutide 47 (21.0) Dulaglutide 16 (7.1) Tirzepatide 12 (5.4) Exenatide 1 (0.5)	Baseline status for comparison of IBD outcomes. non-IBD controls on GLP-1RA (N = 224) for comparison of BMI change.	**IBD outcomes:** Composite of IBD-related hospitalisation, corticosteroid prescription, medication escalation or changes, or IBD related surgery **Metabolic outcomes:** Change in BMI	**IBD outcomes:** no change in IBD exacerbation in the year following GLP-1RA initiation compared with the year before **Metabolic outcomes:** median BMI decrease from 33.5–31.6 kg/m^2^; *p* = <0.01, comparable to non-IBD matched controls		10/10	10/10
St-Pierre et al. (2024) [31]	USA, 2021–2024	Retrospective cohort study	36 IBD patients, non-diabetic UC n = 12 CD n = 24	Semaglutide or tirzepatide	None	**IBD outcomes:** CRP levels, Faecal calprotectin, change in IBD therapy. **Metabolic outcomes:** Changes in BMI and total body weight	**IBD outcomes:** no significant changes in CRP levels, Insufficient sample size for FCP for statistical analysis, n = 6 change in IBD therapy ^4^ **Metabolic outcomes:** BMI significantly decreased from 34 (IQR 31.6–36.2) to 31 (IQR 29–36,1) (*p* < 0.0001), TBW significant decreased by a median of 8.15 (IQR 15.9–2.2) kg (*p* < 0.0001)		4/10	4/10
Gorelik et al. (2024) [32]	Israel, 2005–2021	Population cohort	3737 IBD patients with T2DM	GLP-1RA users (n = 633)	Non-GLP-1RA users (n = 3104)	**IBD outcomes:** Composite of steroid-dependence, initiation of advanced IBD therapy, hospitalisation, surgery, or death **Metabolic outcomes:** Not reported	**IBD outcomes:** Full cohort: aHR ^5^ 0.74, 95% CI: 0.62–0.89 UC: aHR 0.71, 95% CI 0.52–0.96 CD: aHR 0.78, 95% CI 0.62–0.99 patients with obesity: aHR 0.61, 95% CI 0.50–0.77, non-obese patients: aHR 0.94, 95% CI 0.67–1.31		10/10	8/10
Nielsen et al. (2024) [33]	Denmark, 2018–2024	Population cohort	61,927 patients with IBD UC n = 41,191 CD n = 20,736	GLP-1RA (Semaglutide) n = 4430)	Non-GLP-1RA users (n = 57,497)	**IBD outcomes:** Time to paralytic ileus or intestinal obstruction **Metabolic outcomes:** Not reported	**IBD outcomes:** 21 (0.5%) GLP-1RA users vs. 1766 (3.1%) patients non GLP1RA users developed ileus or intestinal obstruction Crude HR was 0.66 (95% CI 0.43–1.01) Adjusted HR was 0.57 (95% CI 0.36–0.88) ^7^		11/11	9/11
Pham et al. (2024) [37]	USA, 2001–2022	Case control	IBD patients on anti-obesity medications (AOMs) n = 36 CD n = 21 UC n = 15	Liraglutide n = 9, Semaglutide n = 10 The remainder on AOMs: Phentermine, Phentermine-topiramate and Naltrexone-bupropion	non-IBD controls (n = 36)	**IBD outcomes:** Any increase/change in IBD medications or objective evidence of endoscopic/radiographic inflammation with increased symptoms, corticosteroid use, hospitalisation, or surgery **Metabolic outcomes:** TBW loss in 12 months between cases and controls	**IBD outcomes:** n = 7, 19.4% experienced IBD flare for all anti-obesity medications ^6^		10/10	10/10
							**IBD-related complication**	N (%)		
							**Crohn’s disease flare**	**6 (28.6)**		
							Corticosteroid used	3		
							Change in IBD therapy	4		
							Hospitalization	2		
							Surgery	0		
							**Ulcerative colitis flare**	**1 (6.7)**		
							Corticosteroid used	1		
							Change in IBD therapy	1		
							Hospitalization	0		
							Hospitalization 0	0		
							**Metabolic outcomes:** Case vs. controls weight loss was −6.9 ± 8.3 and −8.1 ± 7 (*p* = 0.3) No difference in %TBWL between cases and controls and similar frequency of GI side effects between case and control for GLP1RAs			

^1^ Anti-Diabetic therapies include: biguanides, insulin, sulfonylureas, thiazolidinediones, SGLT2 inhibitors, and combination therapy. ^2^ Anti-obesity medications included phentermine-topiramate, bupropion-naltrexone, and orlistat. ^3^ Conventional oral hypoglycemics agents included sulfonylureas, dipeptidyl peptidase 4 inhibitors (DPP4i), sodium glucose co-transporter 2 inhibitors (SGLT2is), thiazolidinediones, or metformin. ^4^ N = 25 had repeat CRP levels—no significant changes (median 3 mg/L for initial & repeat), N = 3 had repeat faecal calprotectin (not enough for statistical analysis), one repeat FCP was 2080 mcg/g, prompting steroid commencement (5 months after initiating semaglutide), and N = 6 had a change in IBD therapy after starting GLP-1 RA (N = 4 had signs of inflammation prior starting GLP-1 RA). ^5^ Adjusted hazard ratio (aHR). ^6^ Outcomes were based for all AOMs, study did not specify IBD outcomes for GLP-1RA only cases. ^7^ After adjusting for age at diagnosis of IBD, sex, steroid use, prior ileus or intestinal obstruction, bowel surgery, diabetes, and type of IBD.

**Table 2 biomedicines-13-01128-t002:** Human in vitro studies on the effect of GLP-1RA on IMIDs.

Author, Year of Publication, Country	Study Population	What Was Studied	Results		
Faurschou, 2013, Denmark [38]	Psoriasis n = 6, (3 males, age (40–64)). 5/6 with no other psoriasis treatment Healthy controls n = 6 (3 males, age (21–287))	GLP1R gene expression ^1^ in skin and blood of pts with psoriasis vs. healthy controls Were GLP1Rs expressed in keratinocytes or immune cells with or without stimulation with TNFα+/− γIFN? IL-17 expression	**Psoriasis pt, n = 6**		**Healthy control, n = 6**
			5/6 expressed GLP1R in affected skin 5/6 had NO GLP1R expression in unaffected skin Numerically higher level of GLP-1R expression in blood		5/6 did not express GLP1R Numerically lower level of GLP-1R expression in blood RNA but *p* = 0.19 (vs. psoriasis patients)
			No GLP1R expression from any of the keratinocyte cell cultures despite stimulation with TNFα and γ IFN: expressed in immune cells IL-17 significantly increased in psoriasis plaques compared with healthy control *p* > 0.03)		
Tao, 2019, China [39]	Rheumatoid arthritis patients n = 12,	To study effect of exenatide on a TNFα-stimulated cell culture of FLS ^2^, including FLS exposed to 10 ng/mL of TNF-α for 24 h Mitochondrial function (via measuring intracellular mitochondrial membrane potential (MMP) ^5^ and cytochrome C oxidase activity) RNA expression via real-time quant PCR of: IL1β, TNFα, IL-6, MMP 3 and 13, and GADPH ^3^. Protein expression of HMG1 ^8^, IL1β, TNFα, IL-6, and MMP 3 and 13 ^4^ also measured by ELISA Protein expression of p65, laminin B1, phosphor-p38MAPK, NOX-4 ^7^, and phosphor-IκBα via Western Blot Oxidative stress via measuring intracellular level of glutathione (GSH) and ROS ^6^ via fluorometric assay NFκB transcriptional activity via luciferase reporter assay	**Inflammatory Variable tested**	**Effect of TNF-α stimulation at 10 ng/mL for 24 h**	**Effect of adding exenatide (in increasing doses, 10 and 20 nM)**
			Mitochondrial function	Decreased MMP and cytochrome C oxidase activity	Increased MMP and cytochrome C oxidase activity *p* < 0.01
			NOX-4 [7] expression and oxidative stress (GSH, ROS)	NOX-4 protein and mRNA expression significantly upregulated. GSH levels reduced, and ROS increased.	Expression of NOX-4 protein and mRNA attenuated *p* < 0.01 Increased levels of GSH and decreased ROS *p* < 0.01
			HMG1 expression	Significantly increased	Attenuation of HMG1 back to normal levels
			IL-1β, IL-6 expression (protein and mRNA), and MMP 3 and 13	All significantly increased	All attenuated with increased doses *p* < 0.01
			p38/MAPK and NF-κB proinflammatory signalling pathways	Increased expression of phosphor-p38 (p-p38), phospho-IκBα ^9^ (p-pIκBα), nuclear p65, and NFκB luciferase activity	Expression of p-p38, p- IκBα, nuclear p65, and NFκB luciferase activity, attenuated with increasing doses *p* < 0.01.
Du, 2019, China [40]	Rheumatoid arthritis patients with joint replacements, n = 10	Study effect of adding lixisenatide (10 and 20 nM) to 10 ng/mL of IL-1β stimulated cell culture of FLS ^2^ Oxidative stress by measuring ROS using fluorometric assay, expression of 4-HNE ^10^ by immunostaining Mitochondrial dysfunction by measuring MMP ^5^ Apoptosis by measuring LDH ^11^ TNFα, IL-6, IL-8, and MMP-1/3/13 ^4^ expression by PCR and ELISA assay Activation of proinflammatory signalling pathways (JNK via Western Blot, AP-1 and p-65/NFκB via luciferase assay)	**Inflammatory Variable tested**	**Effect of IL-1β stimulation at 10 ng/mL for 24 h**	**Effect of adding lixisenatide (in increasing doses, 10 and 20 nM)**
			ROS level	Sig increased	Sig decreased at increased doses *p* < 0.01
			4-HNE expression	Sig increased	Sig decreased at increased doses *p* < 0.01
			MMP	Sig decreased	Sig increased at increased doses *p* < 0.01
			LDH	Sig increased	Sig decreased at increased doses *p* < 0.01
			Expression of inflammatory cytokines and matrix metalloproteinases	Sig increased	Sig decreased at increased doses *p* < 0.01
			Phospho-JNK AP-1 p-65/NFκB	Sig increased	Sig decreased at increased doses *p* < 0.01

^1^ Study could not reliably isolate GLP1R protein by immunohistochemistry ^2^ FLS = fibroblast like synoviocytes, cells within the synovium of RA that take on a pathological apoptosis resistant phenotype which promotes pannus formation. ^3^ GADPH = Glyceraldehyde- 3-phosphate dehydrogenase, ^4^ MMP-1/3/13 = matrix metalloproteinase 3/13. ^5^ MMP = mitochondrial membrane potential ^6^ ROS = Reactive Oxygen Species. ^7^ NOX-4 is involved in the release of superoxide by synoviocytes ^8^ HMG1- regulates expression of pro-inflammatory cytokines. ^9^ IκBα is a transcription factor, where the phosphorylated of form is the active component upon exposure to pro-inflammatory cytokines. Once phosphorylated, IκBα is degraded ‘letting go; of p65 that moves into the nucleus and activates the NF-κB proinflammatory pathway. ^10^ 4HNE = 4 hyroxynonenal, a byproduct of lipid peroxidation, a marker of oxidative stress ^11^ LDH = lactate dehydrogenase, a marker of cell death.

**Table 3 biomedicines-13-01128-t003:** Human clinical studies of GLP-1RAs on IMIDs.

Author (Year)	Study Origin/Year	Study Type	Patient Cohort (N, Mean Age)	Intervention	Comparator	Major Outcomes Reported	Result									JBI Score, No. of ‘Yes’	
																Author 1	Author 2
Nicolau 2023 [41]	Spain	Prospective open-label cohort study	Non-Diabetic Obese Psoriasis patients (N = 20, 45.4 ± 9.7 yrs) 30% had PsA.	Liraglutide 3 mg daily SC for 12 weeks + diet (−500 calories daily) + 150 min aerobic exercise) All continued biologic (16/20) or photo therapy (4/20)	Baseline status	**IMID outcomes:** PASI DLQI VAS **Weight outcomes** Weight BMI Waist circumference	PASI (pre/post): 10 ± 2/5 ± 1, *p* < 0.001 DLQI (pre/post): 13 ± 2/6 ± 1, *p* = 0.009 VAS (pre/post): 4.1 ± 0.4/2.3 ± 0.2, *p* = 0.009 Weight (pre/post): 110.1 ± 21/102.5 ± 21.7, *p* = 0.004 BMI (pre/post): 38.9 ± 5.8/36.4 ± 5.6, *p* = 0.003 Waist circ: 110 ± 8.4/107.2 ± 8, *p* = 0.04									6/11	5/11
Lin 2022 [42]	China 2017–2019	RCT Open label	Type 2 Diabetic Psoriasis patients N = 24, total. N = 13 intervention mean age 56.73 ± 8 yrs, N = 11 comparator, mean age 55.23 ± 7.84)	1.8 mg Liraglutide daily SC for 12 weeks Continued conventional oral anti-diabetic meds	Conventional treatment (oral acitretin capsules 30 mg–50 mg/d + calcipotriol ointment) Continued conventional oral anti-diabetic meds	**IMID outcomes:** Mean change in PASI DLQI IL-23/IL-17/TNFα expression in skin **Weight outcomes** Weight BMI Waist circumference	PASI: control/Int: –6.15 ± 3.43/–12.32 ± 10.05, *p* = 0.049 DLQI: control/Int: –8.54 ± 5.33/–18.18 ± 5.86, *p* < 0.001 IL-23/IL-17/TNFα expression in skin of intervention lower than control *p* < 0.05 Weight: control/Int: 1.92 ± 2.33/–4.82 ± 2.04, *p* < 0.001 BMI: control/Int: 0.69 ± 0.84/–1.77 ± 0.73, *p* < 0.001 Waist Circ: Control/Int: 2.23 ± 2.35/–5.05 ± 3.76 *p* < 0.001									9/13	8/13
Ahern 2013 [43]	Ireland 2010–2011	Prospective open-label cohort	(7, 48 (40–58)) Obese and diabetic psoriasis pts, none with PsA	0.6 mg for 2 weeks then 1.2 mg daily of Liraglutide SC for 10 weeks total	Baseline status	**IMID outcomes:** PASI DLQI % of circulating iNKT cells % of TNFα-producing monocytes **Weight outcomes** Weight	PASI (pre/post): 4.8 (2.6–11.4)/3.0 (1.9–7.9), *p* = 0.03) DLQI (pre/post): 6 (3.5–8.9)/2 (1–6.1), *p* = 0.03) %iNKT cells: increased by 37.9% (IQR:18.5–234.6, *p* = 0.03) % of TNFα-producing monocytes: −53% (IQR: 51.4–55.0) *p* = 0.07 Weight (pre/post): 137.8 kg (120–178 kg)/130.1 kg (115–166 kg, *p* = 0.06)									6/11	5/11
Xu 2019 [44]	China 2017–2018	Prospective open-label cohort	(7, 60 ± 8 yrs) Psoriasis patients with type 2 DM, none with PsA	Liraglutide 0.6 mg/1.2 mg/week each then max dose of 1.8 mg daily for 12 weeks No concurrent treatment with topical therapy, oral IS or biologic therapy	Baseline status	**IMID outcomes** PASI DLQI **Weight outcomes** BMI Abdo circumference	PASI (pre/post): 15.7 (1.5–31.3)/2.0 (0.3–8.7), (*p* = 0.03) DLQI (pre/post): 22 (8–27)/4 (0–10), (*p* = 0.001) Histopathology: significant reduction in neutrophils, Munro micro abscess and thickness of epidermal layer BMI (pre/post): 23 ± 4 kg/m^2^/21 ± 3 kgm^2^ (*p* < 0.01) Waist circ (pre/post): 87 ± 9 cm/83 ± 1 cm (*p* < 0.04)									6/11	7/11
Buysschaert 2014 [45]	Belgium 2011–2012	Prospective open-label cohort	(7, 56 ± 8 yrs) Psoriasis patients with type 2 DM, having failed systemic and topical agents previously None with PsA	Exenatide n = 1 Liraglutide n = 6 At 7 and 18 weeks Dose not described	Baseline status	**IMID outcomes** PASI DLQI Histological activity IL-17 expression γδ T cells **Weight outcomes** BMI	PASI (pre/post 18 wks.): 12 ± 5.9/9.2 ± 6.4 (*p* = 0.04) Epidermis and corneum layer thickness decreased (pre/post): 0.47 ± 0.12 mm/0.40 ± 0.15 mm (*p* = 0.06). No change in dermis infiltrate, granular layer, or the presence of Munro micro-abscesses Decreased γδ T cells (pre/post): 6.7 ± 4.5%/2.7 ± 3.8%, with change in γδ T cells correlating with change in PASI score. (r = 0.894, *p* = 0.007) IL-17 expression decreased numerically but not statistically BMI (pre/post) 32.0 ± 10.1 kg/m^2^/30.6 ± 9.1 kg/m^2^									4/11	4/11
Hogan 2011 [46]	Ireland	Case series	n = 3 (index +2 additional). Mean age = 52.3 Obese type 2 DM with psoriasis on no current treatment	Patient 1 = exenatide (2 months), then Liraglutide 9 months Patient 2 = Liraglutide, 6 weeks Patient 3 = Liraglutide, 6 weeks Dose not reported	Baseline	**IMID outcomes** PASI % iNKT cells skin and blood **Weight outcomes** BMI	PASI (pre/post) Patient 1: >15/10.5 Patient 2: 13.2/10.8 Patient 3: 4.8/3.8 Skin %iNKT cells (pre/post) Patient 1: NR Patient 2: 2.16/0.07 Patient 3: 0.32/0 Blood %iNKT cells (pre/post) Patient 1: NR Patient 2: 0.15/0.6 Patient 3: 0.16/0.57 BMI (pre/post) Patient 1: 37 kg/m^2^/NR Patient 2: 48.0/46.5 kg/m^2^ Patient 3: 43/41.1 kg/m^2^ GLP-1R expressed on iNKT cells GLP-1 induced a dose-dependent inhibition of iNKT cell cytokine secretion (γIFN, IL-2), but not cytolytic degranulation in vitro Increased cAMP—activates CREB transcription factor for IL-10 production									7/10	7/10
Faurschau 2015 [47]	Denmark	Double blind RCT 1:1	(20, 48 ± 12 vs. 54 ± 14; liraglutide vs. placebo) Obese BMI (≥ 25 kg/m^2^), glucose-tolerant pts with plaque psoriasis PASI ≥ 8 PsA patients excluded	Liraglutide n = 11 (0.6, 1.2, 1.8 mg, increasing by 1 week, total 8 weeks No concomitant treatment: n = 4 Topical tx 1-2x daily: n = 6 Systemic tx: n = 0 No change in psoriasis treatment for the prior 3 months	Placebo n = 9, total 8 weeks No concomitant treatment: n = 6 Topical tx 1-2x daily: n = 3 Systemic tx: n = 1 (adalimumab) No change in psoriasis treatment for the prior 3 months	**IMID outcomes** дPASI (mean): from baseline for each group and between groups дDLQI (mean) from baseline and between groups **Weight outcomes** Д weight (mean)	**Outcome**			**Placebo**		**Liraglutide**		***p* value**		13/13	13/13
							дPASI			−1.3 ± 2.4		−2.6 ± 2.1		0.228			
							дPASI from baseline			*p* = 0.14		*p* = 0.0026		-			
							дDLQI: no difference between groups and compared to baseline for both placebo and liraglutide groups.										
							**Outcome**			**Placebo**		**Liraglutide**		***p* value**			
							Д weight			−1.5 ± 2.7		−4.7 ± 2.5		0.014			
Petkovic-Dabic 2025 [48]	Bosnia and Herzegovina 2024	Open label RCT	(31, 58.6 ± 8.04 57.4 ± 13.02; semaglutide vs. placebo) Moderate to severe plaque psoriasis pts (PASI score ≥ 10) with obesity (≥30 kg/m^2^) diagnosed for ≥6 months No PsA patients	N = 15 Semaglutide up to 1 mg with metformin for 12 weeks Anti-obesogenic/diabetic diet Topical Keratolytic therapy and salicylic acid No systemic therapy for prior 3 months for psoriasis, or phototherapy	N = 16 Metformin only Anti-obesogenic/diabetic diet Topical Keratolytic therapy and salicylic acid No systemic therapy for prior 3 months for psoriasis or phototherapy	д **IMID outcomes** PASI (mean): from baseline for each group дDLQI (mean) from baseline for each group дCytokine expression and inflammatory markers (CRP, IL-1β, IL-6, and IL-23) **Weight outcomes** Д BMI	**Outcome**	**Semaglutide**			**Control**					9/13	8/13
							**Med PASI (IQR)**	Wk 0	Wk 12	P	Wk 0		Wk 12		P		
								21 (19.8)	10 (6)	0.002	20.6 (8.9)		15.9 (8.7)		0.03		
							**Med DLQI (IQR)**	14 (5)	4 (4)	0.002	10.1 (4.3)		8.1 (4.8)		0.007		
							**CRP level mg/L**	3.8 (3.1)	1.9 (1.4)	0.01	9.6 ± 10.7		7.6 ± 8.3		0.5		
							**IL-6 pg/mL**	3.5 (2.3)	2.8 (1.1)	0.05	5.6 (12.2)		2.3 (3.6)		0.1		
							**BMI**	33.04 ± 2.7	30.7 ± 3.8	0.001	36 ± 7.9		34.9 ± 7.9		<0.001		
							No changes in other cytokine expression. IL-17 not detected at all.										

PASI = Psoriasis Activity Severity Index, DLQI = Dermatology-Related Quality of Life Index, BMI= Body Mass Index VAS = Visual analogue scale (of pain), PsA = Psoriatic arthritis Int = Intervention. iNKT—invariant natural killer cells, % of TNFα-producing monocytes—after stimulation by lipopolysaccharide and measured by flow cytometry after being labelled by mAb to intracellular cytokines. BMI = body mass index, NR = Not reported, T2DM = Type 2 Diabetes Mellitus.

## Data Availability

Data are available on request.

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
