# Peer review of "GLP-1R Agonists and Their Therapeutic Potential in Inflammatory Bowel Disease and Other Immune-Mediated Inflammatory Diseases, a Systematic Review of the Literature"

_biomedicines, 2025, doi:10.3390/biomedicines13051128_

Round 1
Reviewer 1 Report
Comments and Suggestions for Authors
Previous studies have demonstrated the pivotal role of GLP-1R agonists in obesity. This review, however, focuses on the therapeutic potential of GLP-1R agonists in inflammatory bowel disease (IBD). The article is comprehensive and holds potential value for clinical applications. I only have a few minor suggestions for the authors to consider:
Detailed Comments:
- Lines 163-167: Please ensure consistent font size throughout the text.
- Line 210: Bacterial species such as Lactobacillus reuteri should be italicized—please check the entire manuscript for consistency.
- The authors should carefully review the citation format of the references.
- In addition to discussing the role of GLP-1R agonists in IBD, the authors also summarize their effects in non-IBD immune-mediated inflammatory diseases (IMIDs). Therefore, I suggest revising the title to encompass the full scope of the article’s content.
Author Response
1. Lines 163-167: Please ensure consistent font size throughout the text.
Thank you, we have made amendments throughout the text with yellow highlights particularly for lines 163 -167.
2. Line 210: Bacterial species such as Lactobacillus reuteri should be italicized—please check the entire manuscript for consistency.
Thank you for this comment, all bacterial species have been correctly italicized including Supplementary Table 1.
3. The authors should carefully review the citation format of the references.
Thank you, we have reformatted the references as requested by the editor to include all the authors or at least 10.
4. In addition to discussing the role of GLP-1R agonists in IBD, the authors also summarize their effects in non-IBD immune-mediated inflammatory diseases (IMIDs). Therefore, I suggest revising the title to encompass the full scope of the article’s content
Thank you for this comment, we have amended the title to " GLP-1R agonists and their therapeutic potential in Inflammatory Bowel Disease and other immune-mediated inflammatory diseases, a systematic review of the literature"
Reviewer 2 Report
Comments and Suggestions for Authors
Dear Authors and Editors
Review entitled: GLP-1R agonists and their therapeutic potential in Inflammatory Bowel Disease, a systematic review of the literature, raises a very important issue of the effect of drugs used in obesity on another disease entity with a planned basis, which is IBD. Treatment of obesity with GLP-1 is an important and increasingly common method, I assume that it will become more and more common. Therefore, it is worth considering all the shadows and lights of pharmacological treatment of excess adipose tissue and complications associated with it. However, the manuscript requires refinement in many respects and does not fully focus on what has been described as the subject of the review.
1. The genus and species name of the bacteria is written in italics
2. The Supplementary Table should be enriched with a more detailed description of what breed of animals and in what quantities. Additionally, in the supplementary materials there is a description of "Table 1", it should be called "Supplementary Table 1" as in the text (line 133). The title of this table is missing.
3. In the supplementary table, not all cited references were discussed in the same way, I suggest modifying the table to make it more readable.
4. The title of Tables 1, 2 and 3 is missing.
5. There is no appropriate description of abbreviations under the tables.
6. Placing a table in a table only introduces chaos. Can this be solved differently?
7. It seems that Table 3 should be added as supplementary because it is the longest and gives an artificial volume to the manuscript.
8. Tables 2 and 3 and subsection 3.3 are unnecessary because they contribute nothing to the topic of IBD. It seems that the aspect of immunomodulatory properties of GLP-1 in relation to immunological diseases other than IBD can be discussed with the conclusion that there is a premise that they can be applied to IBD, but a review of studies in such an extensive form that are not related to the topic is unnecessary. In the discussion lines 520 and 521 you suggest: “To the best of our knowledge, this systematic review is the first to examine the available published scientific evidence on the therapeutic potential of GLP-1RAs in treating IBD”. Please stick to that.
Author Response
1.The genus and species name of the bacteria is written in italics
Thank you for this comment. As per reviewer #1's note, we have made this change throughout the manuscript to ensure all species names of bacteria is written in italics
2. The Supplementary Table should be enriched with a more detailed description of what breed of animals and in what quantities. Additionally, in the supplementary materials there is a description of "Table 1", it should be called "Supplementary Table 1" as in the text (line 133). The title of this table is missing.
Thank you for this comment. We have amended Supplementary Table 1 as suggested including line 133. We have added the title of the table, "Animal and in-vitro studies of GLP-RAs on experimental colitis."
3. In the supplementary table, not all cited references were discussed in the same way, I suggest modifying the table to make it more readable.
Thank you for your comment. The supplementary table was particularly challenging to construct due to the complexity and density of scientific information that required summarisation. We followed this format because it was the most effective way to present the essential information without redundancy and excessive detail. We have chosen not to modify this table any further so as not to complicate the narrative that follows the main text.
4 The title of Tables 1, 2 and 3 is missing.
Thank you for this comment, this has been amended
5. There is no appropriate description of abbreviations under the tables.
Thank you for this comment however there are extensive footnotes under each of the tables that define the scientific abbreviations.
6. Placing a table in a table only introduces chaos. Can this be solved differently?
Thank you for your comment. We've lightened the borders to make the nested table less obvious. This was the best way to simplify complex scientific data.
7. It seems that Table 3 should be added as supplementary because it is the longest and gives an artificial volume to the manuscript
Thank you for your comment. The information in Table 3 is integral to the manuscript’s message (Section 3). It underscores that GLP-1RAs have predominantly been extensively researched in non-IBD IMIDs, as opposed to IBD. These studies serve as crucial examples for similar research that should be conducted in IBD patients but has not yet been undertaken.
8. Tables 2 and 3 and subsection 3.3 are unnecessary because they contribute nothing to the topic of IBD. It seems that the aspect of immunomodulatory properties of GLP-1 in relation to immunological diseases other than IBD can be discussed with the conclusion that there is a premise that they can be applied to IBD, but a review of studies in such an extensive form that are not related to the topic is unnecessary. In the discussion lines 520 and 521 you suggest: “To the best of our knowledge, this systematic review is the first to examine the available published scientific evidence on the therapeutic potential of GLP-1RAs in treating IBD”. Please stick to that.
Thank you for this comment. We have updated the title as suggested by Reviewer #2 to indicate that this systematic review includes the topic of non-IBD IMIDs. Although this increases the length of the manuscript, it is important to include lessons from the non-IBD IMID literature that are applicable to designing future studies in IBD patients. The earliest clinical evidence regarding the anti-inflammatory role of GLP-1RAs was observed in non-IBD IMID patients. Additionally, the only randomised controlled trials involving GLP-1RAs have been conducted with non-IBD IMID patients, and there is currently no randomised controlled evidence in IBD patients for the use of GLP-1RAs to control disease. Accordingly, Table 2, Table 3, and subsection 3.3 are fundamental in establishing a comprehensive evidence base for this systematic review. We have also provided an amendment in the discussion as to why we had to present the evidence in non-IBD IMIDs. In lines 514 to 517 we state the following:
Additionally, we reviewed the evidence regarding the utilization of GLP-1 receptor agonists (GLP-1RAs) in treating other immune-mediated inflammatory diseases (IMIDs). This was done to evaluate their therapeutic potential in the management of inflammatory conditions and to contextualize their application.
Reviewer 3 Report
Comments and Suggestions for Authors
This systematic review, conducted by Lena Thin and Wei Ling Teh, explores the potential therapeutic role of GLP-1 receptor agonists (GLP-1RAs) in treating inflammatory bowel disease (IBD). The authors searched multiple databases, including PubMed/Medline, Cochrane Clinical Trials, and EMBASE, using a predefined search strategy and eligibility criteria to examine evidence regarding GLP-1RAs use in both IBD and non-IBD immune-mediated inflammatory disease (IMID) patients.
The authors noted that while animal studies demonstrate GLP-1RAs potential to reduce inflammation and improve colitis, limited clinical studies have investigated their effectiveness in IBD patients. Based on available data, the authors suggests that GLP-1RAs may improve IBD outcomes, including reduced surgery, hospitalization, and corticosteroid use. The authors conclude that although GLP-1RAs show promise, further research, particularly prospective studies and randomized controlled trials (RCTs), is necessary to determine their effectiveness in managing IBD.
The review is well-written in a scientific manner and covers most aspects required for a good review. However, it would be nice if could add the potential side effects and safety concerns associated with using GLP-1RAs in IBD patients. In addition, I noticed some typographical and grammatical errors throughout the text, which should be corrected before publication. In my opinion, this review will be of interest to readers and is suitable for publication.
Author Response
It would be nice if could add the potential side effects and safety concerns associated with using GLP-1RAs in IBD patients. In addition, I noticed some typographical and grammatical errors throughout the text, which should be corrected before publication. In my opinion, this review will be of interest to readers and is suitable for publication.
Thank you for your comments on our review. We acknowledge that the safety and tolerability section was relatively underrepresented in this systematic review, as the focus was primarily on the anti-inflammatory actions of GLP-1RAs. We were mindful of extending the manuscript length. Nonetheless, we have briefly outlined the overall findings of safety and tolerability in each of the IBD sections (lines 377-384) and non-IBD IMID sections (lines 483-485). Typographical and grammatical corrections have been made.
Round 2
Reviewer 2 Report
Comments and Suggestions for Authors
The authors have corrected most of the comments. Although they look better, I have the impression that the tables are still unfinished and could be improved by modifying the content. However, if the Editor deems it appropriate, the manuscript can be published.
Author Response
The authors have corrected most of the comments. Although they look better, I have the impression that the tables are still unfinished and could be improved by modifying the content. However, if the Editor deems it appropriate, the manuscript can be published.
Thank you for your review and comments. We have made a slight change to the table by further expanding some abbreviations, such as AA=acetic acid, Lig=liraglutide, Dula=dulaglutide, NE=norepinephrine. However, all scientific abbreviations cannot be defined exhaustively as some gene expression nomenclature is inherently expressed in specific terms that cannot be further defined. The overall format and content have not been changed as it is complete in its current form and extending the length may complicate the information presented. Please see the attached amended supplementary Table 1